# Nafamostat–Interferon-α Combination Suppresses SARS-CoV-2 Infection In Vitro and In Vivo by Cooperatively Targeting Host TMPRSS2

**DOI:** 10.3390/v13091768

**Published:** 2021-09-04

**Authors:** Aleksandr Ianevski, Rouan Yao, Hilde Lysvand, Gunnveig Grødeland, Nicolas Legrand, Valentyn Oksenych, Eva Zusinaite, Tanel Tenson, Magnar Bjørås, Denis E. Kainov

**Affiliations:** 1Department of Clinical and Molecular Medicine (IKOM), Norwegian University of Science and Technology, 7028 Trondheim, Norway; rouan.yao@ntnu.no (R.Y.); hilde.lysvand@ntnu.no (H.L.); valentyn.oksenych@ntnu.no (V.O.); magnar.bjoras@ntnu.no (M.B.); 2Research Institute of Internal Medicine, Oslo University Hospital Rikshospitalet, 0372 Oslo, Norway; gunnveig.grodeland@medisin.uio.no; 3Institute of Clinical Medicine (KlinMed), University of Oslo, 0318 Oslo, Norway; 4Section of Clinical Immunology and Infectious Diseases, Oslo University Hospital Rikshospitalet, 0372 Oslo, Norway; 5Oncodesign, 25 Avenue du Québec, 91140 Villebon Sur Yvette, France; nlegrand@oncodesign.com; 6Institute of Technology, University of Tartu, 50411 Tartu, Estonia; eva.zusinaite@ut.ee (E.Z.); tanel.tenson@ut.ee (T.T.); 7Institute for Molecular Medicine Finland, FIMM, University of Helsinki, 00014 Helsinki, Finland

**Keywords:** SARS-CoV-2, interferon-alpha, nafamostat, antiviral drug combination, broad-spectrum antivirals

## Abstract

SARS-CoV-2 and its vaccine/immune-escaping variants continue to pose a serious threat to public health due to a paucity of effective, rapidly deployable, and widely available treatments. Here, we address these challenges by combining Pegasys (IFNα) and nafamostat to effectively suppress SARS-CoV-2 infection in cell culture and hamsters. Our results indicate that Serpin E1 is an important mediator of the antiviral activity of IFNα and that both Serpin E1 and nafamostat can target the same cellular factor TMPRSS2, which plays a critical role in viral replication. The low doses of the drugs in combination may have several clinical advantages, including fewer adverse events and improved patient outcome. Thus, our study may provide a proactive solution for the ongoing pandemic and potential future coronavirus outbreaks, which is still urgently required in many parts of the world.

## 1. Introduction

Over the past 10 years, there have been four major viral epidemics/pandemics for which the world was unprepared. The current pandemic concerns the SARS-CoV-2 virus, which has infected over 281 million people globally, killed over 3.9 million, devastated economies, and caused other unfathomable hardships. The challenge addressed in this study is a tragic lack of effective antiviral drugs that can be deployed to treat SARS-CoV-2 infection.

So far, many monotherapies have been tested, but have been shown to have limited efficacy against COVID-19 [1]. By contrast, combinational therapies are emerging as a useful tool to treat SARS-CoV-2 infection [2,3,4,5,6,7,8,9]. Synergistic combinational therapies can achieve the same or better efficacy while requiring lower dosage compared to monotherapies, therefore, inducing fewer and milder adverse effects. Additionally, antiviral combinations could target emerging SARS-CoV-2 variants and prevent the development of strains resistant to monotherapies.

Our recent studies have highlighted the synergism of several compounds against SARS-CoV-2 in human lung epithelial Calu-3 cells, with IFNα–remdesivir and camostat–remdesivir combinations having the highest synergy scores [2,10,11]. In these studies, we were also able to show that camostat–remdesivir was effective against SARS-CoV-2 infection in human lung organoids, as well as that IFNα–remdesivir was effective in both human lung organoids and Syrian hamsters [10,11]. Furthermore, other studies have shown that combinations of IFNα with lopinavir–ritonavir–ribavirin as well as nafamostat with favipiravir are effective for treatment of patients infected with SARS-CoV-2 [12,13].

Nafamostat, a structural analogue of camostat, is a repurposed drug which was originally approved as a short-acting anticoagulant and is also used for the treatment of pancreatitis [14]. It is currently in clinical trials for treatment of COVID-19 (NCT04623021, NCT04473053, NCT04390594, and NCT04483960). IFNαs and its pegylated forms are also repurposed drugs which have been shown to be effective for COVID-19 patients [15,16]. Because previous studies have demonstrated the high therapeutic potential of both nafamostat and IFNα as separate antiviral treatments, we hypothesized that IFNα–nafamostat combinational therapy may represent an even more practical therapeutic option against SARS-CoV-2 infection.

## 2. Materials and Methods

### 2.1. Drugs, Viruses, Cells, and Hamsters

Pegasys (cat. # EMEA/H/C/000395, 008767) was purchased from local pharmacy in syringes for subcutaneous injection (135 µg IFNα2a/0.5 mL each). Lyophilized IFNα2a (cat. # 11343504; ImmunoTools, Friesoythe, Germany) was dissolved in sterile deionized water to obtain 200 μg/mL concentrations. Camostat mesylate (cat. # 16018, Cayman Chemicals, Ann Arbor, MI, USA), nafamostat mesylate (cat. # 14837, Cayman Chemicals) and tiplaxtinin (cat. # 393105-53-8, MedChemExpress, Monmouth Junction, NJ, USA) were dissolved in dimethyl sulfoxide (DMSO; Sigma-Aldrich, St. Louis, MO, USA) to obtain 10 mM stock solutions.

Recombinant mCherry-expressing SARS-CoV-2 (SARS-CoV-2-mCherry), and wild type human SARS-CoV-2 strains were provided by Prof. Andres Merits or the European Virus Archive global (EVAg) and propagated in Vero E6 or Vero E6/TMPRSS2 cells. To quantitate the production of infectious virions, we titered the viruses using plaque assays or ELISA.

The propagation of human non-small-cell lung cancer Calu-3 have been described in our previous studies [11,17].

Thirty-five 6-week-old healthy female Syrian hamsters were obtained from Janvier Labs. The animals were maintained in pathogen free health status according to the FELASA guidelines. The animals were individually identified and were maintained in housing rooms under controlled environmental conditions: temperature: 21 ± 2 °C, humidity 55 ± 10%, photoperiod (12 h light/12 h dark), H14 filtered air, minimum of 12 air exchanges per hour with no recirculation. Each cage was labeled with a specific code. Animal enclosures provided sterile and adequate space with bedding material, food and water, as well as environmental and social enrichment (group housing) as described below: IsoRat900N biocontainment system (Techniplast, Decines-Charpieu, France), poplar bedding (Select fine, Safe, Augy, France), A04 SP-10 diet (Safe, France), tap water, environmental enrichment, tunnel, and wood sticks.

### 2.2. Drug Testing and Drug Sensitivity Quantification

Approximately 4 × 10^4^ Calu-3 cells were seeded per well in 96-well plates. The cells were grown for 24 h in DMEM-F12, supplemented with 10% FBS and Pen–Strep. The medium was then replaced with DMEM-F12 containing 0.2% BSA, Pen–Strep, and the compounds in 5-fold dilutions at 7 different concentrations. No compounds, but the vehicle, were added to the control wells. The cells were infected with SARS-CoV-2-mCherry strains at a moi of 0.1 or mock. After 48 h drug efficacy on SARS-CoV-2-mCherry infected cells was measured on PFA- or acetone-fixed cells with fluorescence. After 72 h of infection, a CellTiter-Glo (CTG) assay was performed to measure cell viability.

The half-maximal cytotoxic concentration (CC_50_) for each compound was calculated based on viability/death curves obtained on mock-infected cells after non-linear regression analysis with a variable slope using GraphPad Prism software version 7.0a. The half-maximal effective concentrations (EC_50_) were calculated based on the analysis of the viability of infected cells by fitting drug dose–response curves using four-parameter (*4PL*) logistic function *f*(*x*):(1)$$\;f\left(x\right)=A_{min}+\frac{A_{max}-A_{min}}{1+{\left(\frac{x}{m}\right)}^{\lambda }},$$
where *f*(*x*) is a response value at dose *x*, *A_min_* and *A_max_* are the upper and lower asymptotes (minimal and maximal drug effects), *m* is the dose that produces the half-maximal effect (EC_50_ or CC_50_), and *λ* is the steepness (slope) of the curve. The relative effectiveness of the drug was defined as selectivity index (*SI* = CC_50_/EC_50_).

### 2.3. Drug Combination Testing and Synergy Calculations

Calu-3 cells were treated with different concentrations of two drugs and infected with SARS-CoV-2-mCherry (moi 0.1) or mock. After 48 h, cell viability and reporter protein expression were measured. To test whether the drug combinations act synergistically, the observed responses were compared with expected combination responses. The expected responses were calculated based on the ZIP reference model using SynergyFinder version 2 [18,19]. Final synergy scores were quantified as average excess response due to drug interactions (i.e., 10% of cell survival beyond the expected additivity between single drugs represents a synergy score of 10). Additionally, we calculated most synergistic area scores for each drug combination—the most synergistic 3-by-3 dose-window in dose–response matrixes.

### 2.4. Prophylactic Study of Remdesivir, Pegasys and Their Combination against SARS-CoV-2 Infection in Hamsters

Thirty-five animals were weighed and divided into 6 homogenous groups of 5–6 animals. Group 1 (5 non-infected controls) remained unmanipulated. Group 2 received vehicle (10 mL/kg) 3 times by IP route on Day 0 (t-2 h), Day 1 and Day 2. Group 3 received nafamostat (10 mg/kg) 3 times by IP route on Day 0 (t-2 h), Day 1 and Day 2. Group 4 received Pegasys (40 µg/kg) 3 times by IP route on Day 0 (t-2 h), Day 1 and Day 2. Group 5 received a mixture of nafamostat and Pegasys 3 times by IP route on Day 0 (t-2 h), Day 1 and Day 2. Group 6 received a mixture of the nafamostat and Pegasys 3 times by IP route on Day 0 (t-2 h), Day 1 and Day 2 and Tiplaxtinin (3 mg/kg) 3 times by PO route on Day 0 (t-2 h), Day 1 and Day 2. Groups 2–6 received SARS-CoV-2 intranasally. Animal viability, behavior, and clinical parameters were monitored daily. After 3 days animals were deeply anesthetized using a cocktail of 30 mg/kg (0.6 mL/kg) Zoletil and 10 mg/kg (0.5 mL/kg) Xylazine IP. Cervical dislocation followed by thoracotomy was performed before lung collection. The entire left lungs and superior, middle, post-caval, and inferior lobes of right lungs were put in RNAlater tissue storage reagent overnight at 4 °C, then stored at −80 °C until RNA extraction.

### 2.5. Gene Expression Analysis

Total RNA was isolated using RNeasy Plus Mini kit (Qiagen, Hilden, Germany) from lungs of Syrian hamsters. Polyadenylated mRNA was isolated from 250 ng of total RNA with NEBNext Poly(A) mRNA magnetic isolation module. NEBNext Ultra II Directional RNA Library Prep kit from Illumina was used to prepare samples for sequencing. Sequencing was done on NextSeq 500 instrument (set up: single-end 1 × 76 bp + dual index 8 bp) using NextSeq High Output 75 cycle sequencing kit (up to 400 M reads per flow cell). Reads were aligned using the Bowtie 2 software package version 2.4.2 to the NCBI reference sequence for SARS-CoV-2 (NC_045512.2) and to the Mesocricetus auratus MesAur1.0 assembly genome (https://ftp.ensembl.org/pub/release-100/fasta/mesocricetus_auratus/dna/; accessed on 1 June 2021). The number of mapped and unmapped reads that aligned to each gene were obtained with the ‘featureCounts’ function from ‘Rsubread’ R-package version 2.10. The GTF table for the SARS-CoV-2 reference sequence was downloaded from https://ftp.ncbi.nlm.nih.gov/genomes/all/GCF/009/858/895/GCF_009858895.2_SM985889v3/GCF_009858895.2_ASM985889v3_genomic.gtf.gz (accessed on 1 June 2021). The heatmaps were generated using the pheatmap package (https://cran.r-project.org/web/packages/pheatmap/index.html accessed on 1 June 2021) based on log2-transformed or non-transformed profiling data.

RT-PCR was performed using SuperScript™ III One-Step qRT-PCR System kit (commercial kit #1732-020, Life Technologies) with primers ORF1ab_Fw: CCGCAAGGTTCTTCTTCGTAAG, ORF1ab_Rv: TGCTATGTTTAGTGTTCCAGTTTTC, ORF1ab_probe: Hex-AAGGATCAGTGCCAAGCTCGTCGCC-BHQ-1 targeting a region on ORF1ab. RT-qPCR was performed using a Bio-Rad CFX384™and adjoining software. The relative gene expression differences were calculated using β-Actin as control and the results were represented as relative units (RU). Technical triplicates of each sample were performed on the same qPCR plate and non-templates and non-reverse transcriptase samples were analyzed as negative controls. Statistical significance (*p* < 0.05) of the quantitation results was evaluated with t-test. Benjamini–Hochberg method was used to adjust the *p*-values.

## 3. Results and Discussion

To test our hypothesis, we first analyzed toxicity and efficacy of Pegasys (pegylated IFNα) and nafamostat, against mCherry-expressing SARS-CoV-2 [20] in human lung epithelial Calu-3 cells using fluorescence and cell viability assay as readouts as described previously [11]. We observed that both Pegasys and nafamostat reduced SARS-CoV-2-mediated mCherry expression and rescued cells from virus-mediated death (Figure 1a). Interestingly, Pegasys reduced SARS-CoV-2 replication less efficiently than its non-pegylated analogue, whereas nafamostat reduced SARS-CoV-2 replication more efficiently than camostat.

Second, we examined whether Pegasys–nafamostat can inhibit SARS-CoV-2 infection and protect Calu-3 cells from virus-mediated death more efficiently and at lower concentrations than monotherapies. We tested the antiviral efficacy and toxicity of the combination by monitoring SARS-CoV-2-mediated mCherry expression and cell viability. The drug combination was tested in a 6 × 6 dose–response matrix, where five doses of each combination component are combined in a pairwise manner. As a result, we obtained dose–response matrices demonstrating virus inhibition and cell viability (Figure 1b). We plotted synergy distribution maps, showing synergy (higher than expected effect) at each pairwise dose. We calculated average ZIP synergy scores for the whole 6 × 6 dose–response matrices and for most synergistic 3 × 3 dose-regions, summarizing combination synergies into single metrics. We observed strong synergy of the Pegasys–nafamostat combination (ZIP synergy scores: 4.8 (mCherry) and 27.4 (CTG); most synergistic area scores: 13.6 (mCherry) and 36.4 (CTG)). This strong synergy indicates that, both components could be combined in vitro at decreased concentrations to achieve antiviral efficacy comparable to those of individual drugs at high concentrations.

Next, we examined whether Pegasys–nafamostat can affect the replication of SARS-CoV-2 in vivo (Figure 2a). Four groups of six six-week-old female Syrian hamsters were injected IP with 40 μg/kg Pegasys, 10 mg/kg nafamostat, Pegasys-nafamostat combination or vehicle at day 0, 1, and 2 of infection. After 2 h of drug treatment at day 0, the animals received SARS-CoV-2 strain Slovakia/SK-BMC5/2020 intranasally (10^5^ pfu TCID_50_ per animal). A control group of five hamsters remained untreated and uninfected. After 3 days, the animals were anesthetized and euthanized, and lungs were collected. Total RNA was extracted and polyadenylated RNA was sequenced. We found that the drug combination efficiently attenuated virus-mediated expression of *IDO1, CXCL11*, *CXCL5*, *CCL7*, and other genes which encode immune- and neuromodulators (Figure 2c).

We also found that the drug combination efficiently attenuated synthesis of viral RNA (Figure 2c). We validated our results by analyzing expression of viral ORF1ab using RT-qPCR. We found that the antiviral effect of the combination was additive (synergy score: 5.17; Figure 2d), suggesting that this cocktail has high translatability.

Our recent transcriptomics analysis revealed that IFNα activates transcription of many genes including endothelial plasminogen activator inhibitor (*SERPINE1)* in Calu-3 cells and human lung organoids [11]. Serpin E1 inhibits urokinase-type and tissue-type plasminogen activators (uPA and tPA) as well as various membrane-anchored serine proteases including transmembrane protease serine 2 (TMPRSS2) [21]. Interestingly, nafamostat also inhibits TMPRSS2 [22]. Therefore, we hypothesized that the synergy of the Pegasys–nafamostat could be achieved because both drugs target the same host factor, TMPRSS2 (Figure 2d). To investigate this, we treated a group of six hamsters with an inhibitor of Serpin E1, tiplaxtinin (3 mg/kg, PO), Pegasys (40 μg/kg, IP), and nafamostat (10 mg/kg, IP) at day 0, 1, and 2 of infection. After 2 h of drug treatment at day 0, the animals received SARS-CoV-2 intranasally. After 3 days, the animals were anesthetized and euthanized, and lungs were collected. Total RNA was extracted and polyadenylated RNA was sequenced. Tiplaxtinin treatment restored synthesis of viral and several host RNAs to the level of nafamostat alone, and therefore wholly eliminated the effect of Pegasys (Figure 2c). This was confirmed by RT-qPCR. Our results indicate that Serpin E1 is an essential mediator of IFNα activity and that TMPRSS2 can be targeted by several drugs to synergistically suppress viral infection. However, more targeted studies are needed to support the proposed mechanism of synergy.

It was shown that nafamostat inhibits hypercoagulopathy associated with severe COVID-19 [14]. By contrast, IFNα-induced Serpin E1 is a risk factor for thrombosis [23]. Thus, it is possible that the anticoagulative properties of nafamostat could compensate for the adverse effects of IFNα-induced Serpin E1 when administered in combination with each other. Moreover, combination therapy containing lower doses of Pegasys and nafamostat may reduce the likelihood of developing other side effects entirely [24], and thus be useful in treating COVID-19 patients. In addition, the Pegasys–nafamostat combination could be delivered through different administration routes, leading to greater ease of treatment. We believe further development of combination of these two prescription drugs can lead to practical therapeutic options against many viruses for which replication depends on TMPRSS2, including influenza viruses and other coronaviruses [25]. Furthermore, our proof-of-concept study shows that research on potential synergistic antiviral combinations can have significant global impact, by increasing protection of the general population against emerging and re-emerging viral diseases and filling the time between virus identification and vaccine development with life-saving countermeasures.

## Figures and Tables

**Figure 1 viruses-13-01768-f001:**
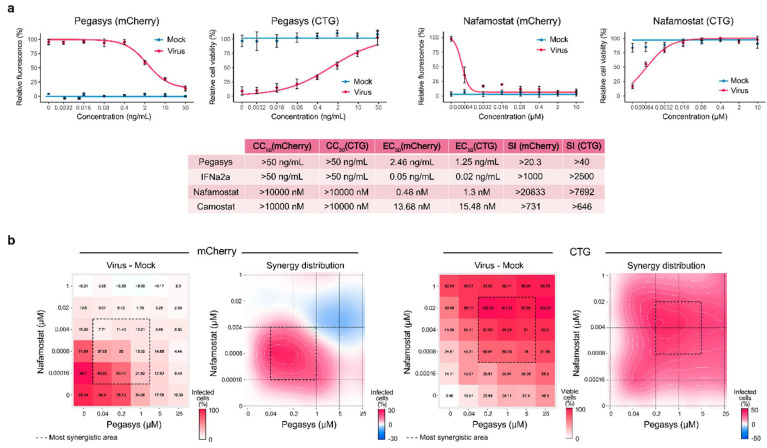
Nafamostat–interferon-alpha combination suppresses SARS-CoV-2 infection in vitro. (**a**) Pegasys and nafamostat attenuate virus-mediated reporter protein expression and rescue Calu-3 cells from SARS-CoV-2-mediated death. Calu-3 cells were treated with increasing concentrations of Pegasys or nafamostat and infected with the SARS-CoV-2-mCherry or mock. After 48 h, the virus-mediated mCherry expression was measured (red curves). After 72 h, viability of virus- and mock-infected cells was determined using a CTG assay (yellow and blue curves, respectively). Mean ± SD; *n* = 3. Toxicity and anti-SARS-CoV-2 activity of Pegasys and nafamostat was quantified and compared to that of IFNα2a and camostat. (**b**) Pegasys and nafamostat combination act synergistically against SARS-CoV-2-mCherry infection in Calu-3 cells. Calu-3 cells were treated with increasing concentrations of IFNα2a, nafamostat, or both drugs and infected with the SARS-CoV-2-mCherry or mock. After 48 h, the virus-mediated mCherry expression was measured (red curves). After 72 h, viability of virus- and mock-infected cells was determined using a CTG assay (yellow and blue curves, respectively). The 6 × 6 dose–response matrices and interaction landscapes of the drug combination was obtained using fluorescence analysis as well as cell viability assay (CTG) on mock-, and SARS-CoV-2-mCherry-infected Calu-3 cells. ZIP synergy score was calculated for the drug combinations. The selectivity for the indicated drug concentrations was calculated (selectivity = efficacy-(100-Toxicity)). ZIP synergy scores were calculated for indicated drug combinations. Representative matrices and interaction landscapes are shown (*n* = 2).

**Figure 2 viruses-13-01768-f002:**
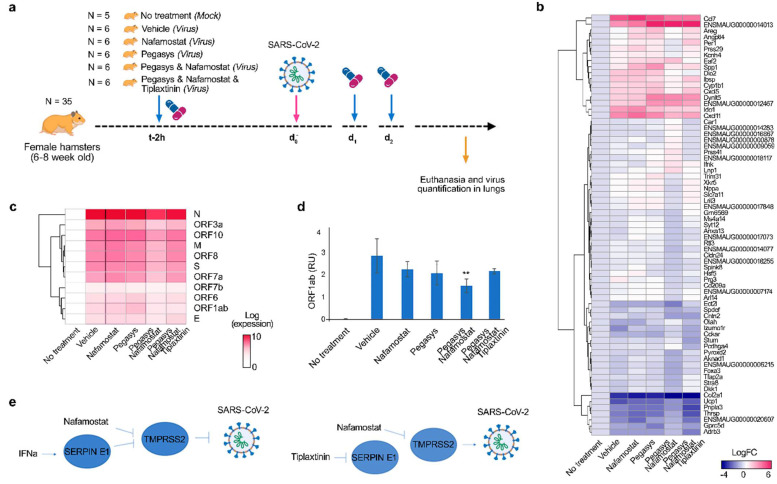
Nafamostat–interferon-alpha combination suppresses SARS-CoV-2 infection in vivo. (**a**) Schematic representation of the experimental setup. (**b**,**c**) Heatmaps of host and viral polyadenylated RNAs affected by treatment. Each cell is colored according to the log2-transformed expression values of the samples, expressed as fold-change relative to the nontreated mock-infected control. Mean, *n* = 6 (except control group, where *n* = 5). (**d**) RT-qPCR analysis of selected viral RNA (**right panel**). Expression of viral RNA was normalized to b-actin control. Mean ± SD, *n* = 6 (except control group, where *n* = 5). Statistically significant differences in viral gene expression between non-treated and treated animals are indicated with asterisks (one-way ANOVA with post hoc Tukey HSD test, **—*p* < 0.3). (**e**) Schematic representation of hypothetical mechanism of action of Pegasys–nafamostat combination on TMPRSS2-dependent SARS-CoV-2 infection. Pegasys–nafamostat combination act cooperatively against SARS-CoV-2 by targeting among other factors TMPRSS2, whereas tiplaxtinin eliminated the effect of Pegasys on TMPRSS2.

## Data Availability

All data generated or analyzed during this study are included in this published article and its Appendix A.

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
