# Peer review of "Nafamostat–Interferon-α Combination Suppresses SARS-CoV-2 Infection In Vitro and In Vivo by Cooperatively Targeting Host TMPRSS2"

_viruses, 2021, doi:10.3390/v13091768_

Round 1

Reviewer 1 Report

In this study authored by Ianevski et. al, the authors have evaluated the potential antiviral activity of Pegasys (IFNα) and nafamostat combination against SARS-CoV-2 viral infection in vitro and in vivo. The authors have done a good description of the rationale behind choosing these combinations of drugs in their introduction section. In vitro and in vivo studies have shown that the Pegasys-nafamostat combination was able to reduce SARS-CoV-2 infection. The authors have also demonstrated that Serpin E1 is a critical mediator of the antiviral activity of IFNα and TMPRSS2 can be targeted by Serpin E1 and nafamostat. TMPRSS2 inhibition is by nafamostat is also previously reported by few other studies –https://doi.org/10.1039/D0SC05064D, https://doi.org/10.1128/AAC.00754-20 and thus this study adds credibility and supports the findings in the field.

Minor corrections:

Line 2, 246: Correct “IFNa” to “IFNα” and in other places wherever applicable

Line 118: Typographical error “IFNαα” must be correct to “IFNα”

Line 144: Please correct the justification of space between the words

Line 183: Correct “IFNa2a” to “IFNα2α” and in other places wherever applicable.

Line 246: Correct “IFNa” to “IFNα” and in other places wherever applicable

Author Response

Many thanks for very positive feedback.

We have now corrected “IFNa, IFNaa, IFNa2a” to “IFNα or ”IFNα2a" wherever applicable.

We have now justified the absence of spaces between the words in the phrase: "the ‘featureCounts’ function from ‘Rsubread’ R-package version 2.10". ‘featureCounts’ is a name of function, whereas ‘Rsubread’ is the name of R-package.

Reviewer 2 Report

This study by Ianevski et al. explores the efficacy of combined therapies of Nafamostat and interferon-alpha in suppressing SARS-CoV-2 infection. The cell-based and animal-based methods used to demonstrate this combination therapy are well-considered and executed, and the synergistic effects of this drug combination support the need for further investigation. 

Some minor comments are summarized below: 

In line 37, the authors highlight the emerging benefit of combinational therapies in combating SARS-CoV-2 infection, using their own work as an example. Combinational therapies are not novel or limited to COVID-19. The efficacy of combined therapies, targeting different mechanisms of action and/or routes of administration is not novel. Numerous examples of effective combinational therapies exist for a variety of indications and should be acknowledged as a precedent.  

In line 91, the authors mention that no compounds were added to the control wells, but are not clear on whether this group received equivalent vehicle control, as done with animal testing.

In line 215, the authors mention that the combination efficiently attenuated the synthesis of viral and host RNA, but do not describe what host RNA was attenuated, or why this would be beneficial.

In Figure 2, Col2a1 (Collagen Type II Alpha 1 Chain) is shown to be significantly attenuated in groups 2-6 relative to Group 1 controls, along with a few others albeit less so. How do the authors account for this observation and discuss their potential relevance? For example, Ucp1 (uncoupling protein 1) is among the downregulated genes and is a mitochondrial proton carrier protein. How might this affect Cell Titer Glo which utilizes mitochondrial ATP production? Several other host genes are upregulated, but their identity is unclear and they are not discussed. It appears that CD7 is upregulated relative to the control group (1). This, of course, is not surprising considering its role in lymphocytes and immune response, but curious as to the identity and relevance of other host genes shown. For example, the Pegasys-treated Groups (4-6) show significant upregulation of ENSMAUG00000014013, which through a literature search is related to 2'-5'-oligoadenylate synthetase 3, an interferon-induced, dsRNA-activated enzyme that plays a role in innate antiviral immune response. There are also several genes showing differential expression between groups 4-6 which could shed light on drug synergy and attenuation by tiplaxtinin.

Figure 2 also includes a schematic mechanism of action, including Nafamostat and SERPIN E1 inhibition of TMPRSS2. What happens in wild-type VeroE6 cells, which have been shown to have limited TMPRSS2 expression? What about other proteases involved in viral infection, replication (e.g. furin, cathepsins), or perhaps host-mediated responses?

In line 233, the authors reference their previous studies of Serpin E1 induction by IFNa, and suggest that Serpin E1 is responsible for the synergistic effects of Pegasys-Nafamostat through combined inhibition of TMPRSS2. Inclusion of Serpin E1 in their animal RNA analysis, showing Serpin E1 induction in the Pegasys-treated groups would help support this. More directly, Serpin E1 knockdown or knockout in Calu-3 would further provide molecular evidence supporting its role in the observed synergy. Likewise, similar cell studies in cells lacking TMPRSS2 (either by knockdown or in cells lacking TMPRSS2), showing no effect by either drug would support this. Furthermore, while tiplaxtinin is considered a selective PAI-1 inhibitor, it is conceivable that inhibition of other targets may be contributing to the reduction of efficacy in the Pegasys-treated group.  Similar attenuation by other PAI-1 inhibitors, such as MDI-2268 would strengthen this conclusion.

Overall, this manuscript is timely, well-written, and clearly presented. This study provides evidence supporting the efficacy of combination therapy using Pegasys (PEG-IFNa-2a) and Nafamostat in the treatment of COVID-19. The cell-based and animal-based methods used to demonstrate this combination therapy are well-considered and executed, and the synergistic effects of this drug combination support the need for further investigation and consideration for clinical trials beyond their single drug investigations. The mechanism of the observed synergy; however, seems somewhat unclear based on the data. While the role of Pegasys-induced Serpin E1-mediated inhibition of Nafamostat is conceivable, more targeted studies are needed to clearly demonstrate the proposed mechanism of synergy. 

Author Response

Line 37. We have now acknowledged other studies: "By contrast, combinational therapies are emerging as a useful tool to treat SARS-CoV-2 infection [2-9]. "

Line 91, The cells in control wells received equivalent vehicle control. "No compounds, but vehicle, were added to the control wells."

Line 215. We have now provided examples of host mRNAs, which were attenuated by drug combination: "We found that the drug combination efficiently attenuated virus-activated expression of IDO1, CXCL11, CXCL5, CCL7, and other genes which encode immune- and neuromodulators (Fig. 2c)." Supplementary information contains mRNA-seq data which allows further investigations.  

Fig. 2b. Questions related to host RNA-seq  are extremely important but very difficult to address using whole-lung RNA seq because different types of lung cells respond differently to SARS-CoV-2 infection and/or drug treatments. Our previous RNA-seq experiments utilizing mock/SARS-CoV-2 infected vehicle/drug-treated human lung epithelial Calu-3 cells and lung organoids confirmed the impact of some (CCL7) but not other genes (such as USP1) (https://www.biorxiv.org/content/10.1101/2021.01.05.425331v5). Single-cell RNA-seq could also provide answers to the questions, but is very expensive to perform and goes beyond our current study. 

Fig. 2e. We agree with the comment. SARS-CoV-2 replication relies on many different host factors including TMPRSS2. IFNa induces expression of many different ISGs including SERPINE1 which inhibits TMPRSS2 and limits SARS-CoV-2 replication. Nafamostat inhibits many different host proteases apart from TMPRSS2, including prothrombin, coagulation factor X and XII, Trypsin-1, and Kallikrein-1 (https://go.drugbank.com/drugs/DB12598) and limit SARS-CoV-2 infection. Indeed,  IFNa-nafamostat combination have multiple modes of antiviral actions in SARS-CoV-2 targeted cells, including Vero-E6. We now corrected the legend of Fig. 2e: "(e) Schematic representation of hypothetical mechanism of action of Pegasys-nafamostat combination on TMPRSS2-dependent SARS-CoV-2 infection. Pegasys-nafamostat combination act cooperatively against SARS-CoV-2 by targeting among other factors TMPRSS2, whereas tiplaxtinin eliminated the effect of Pegasys on TMPRSS2." 

Line 233. We included RNA seq data and graph showing expression levels of SERPINE1 in the supplementary data. The graph shows that IFNa-nafamostat combination attenuates expression of SERPINE1 in lungs of infected hamsters.  

The roles of SERPINE1 and TMPRSS2 in SARS-CoV-2 infection are also supported by other studies (https://www.ncbi.nlm.nih.gov/pmc/articles/PMC7997307/ https://doi.org/10.1128/AAC.00754-20, https://www.pnas.org/content/118/1/e2021450118). Thus,  our study adds credibility and supports the findings in the field.

Conclusions. We agree that more targeted studies are needed to clearly demonstrate the proposed mechanism of synergy. We added this statement to the results and discussion section: "However, more targeted studies are needed to support the proposed mechanism of synergy".